# Intraspecific Evaluation of Phenotypic Variations of *Caryopteris incana* (Thunb. ex Houtt.) Miq. in Western Kyushu, Japan

**DOI:** 10.3390/plants14121840

**Published:** 2025-06-15

**Authors:** Masaya Ando, Takanori Kuronuma, Hitoshi Watanabe

**Affiliations:** 1Kyoto Herbal Garden, Takeda Pharmaceutical Company, Limited, Kyoto City 606-8134, Kyoto Prefecture, Japan; masaya.ando@takeda.com; 2Center for Environment, Health and Field Sciences, Chiba University, Kashiwa City 277-0882, Chiba Prefecture, Japan; t.kuronuma@chiba-u.jp

**Keywords:** phenotypic variation, geographic variation, principal component analysis, intraspecific evaluation, *Caryopteris incana*

## Abstract

*Caryopteris incana* (Thunb. ex Houtt.) Miq., an endangered species native to western Kyushu, Japan, forms locally isolated populations. In our previous studies, we conducted a distribution survey of 109 populations across four regions and performed phylogenetic analyses using chloroplast DNA to clarify their genetic structure and diversity. The primary objective of this study was to clearly compare and evaluate phenotypic differences that are considered to be genetically fixed among adaptive traits evolved in response to the native environments of each region. We evaluated morphological characteristics—such as floral parts, leaves, and plant form—under uniform cultivation conditions using seeds collected from native populations. As a result, region-specific phenotypes were identified, and diverse variations were observed both among and within regions. Furthermore, the presence of phenotypes with high coefficients of variation even within the same region suggests the existence of diverse gene pools suitable for various applications. This study not only contributes to understanding the regional adaptive strategies of *C. incana*, but also provides fundamental data for developing conservation plans for this endangered species. Additionally, the phenotypic information obtained is expected to be useful for breeding horticultural varieties and for setting future conservation priorities.

## 1. Introduction

*Caryopteris incana* (Thunb. ex Houtt.) Miq. is a perennial herb or shrub native to China, Taiwan, the Korean Peninsula, and Japan (Figure 1). The plants grow from 30 cm to over 1 m tall and are erect and stoloniferous. The ovate leaves are opposite, gray-green, coarsely serrated, and strongly aromatic [1]. Aggregate inflorescences grow from the leaf axils at the top of the stem and bear fragrant blue-purple flowers. Inflorescences are arranged in clusters on the stem and bloom from the lower to the upper part of the flower. A short-day plant, it flowers in the fall and remains in bloom for 1–2 months before the upper leaves wither back. In early winter, a few small leaves remain in rosettes at the base of the plant and over winter. Horticultural uses include funeral flower and floral arrangements, as well as planting in gardens. In particular, *Caryopteris × clandonensis* A. Simmonds (hereinafter referred to as the Caryopteris hybrid), a hybrid with *C. incana* as the seed parent and *C. mongholica* Bunge as the pollen parent, has many hardy garden varieties. The cold-tolerant varieties are used as garden plants and cut flowers mainly in Europe and the United States [2]. However, the only horticultural varieties of *C. incana* f. *candida*, C.K. Schneid. and *C. incana* f. *rosea* Sugim., that are currently available are those with three flower colors: blue-violet, white (f. *candida* C.K. Schneid.), and peach (f. *rosea* Sugim.), and few varieties have been produced that are distinguished by their flower color traits. Few cultivars that can be distinguished by their herbaceous form have been produced. Therefore, phenotypic information on wild species is needed as breeding material to diversify horticultural varieties of *C. incana* and the Caryopteris hybrid. In addition to these horticultural uses, the plant body of *C. incana* contains several phenylpropanoid glycosides, including incanoside, which is used as a folk medicine for whooping cough, menstrual irregularities, eczema, and rheumatic pain in China [1,3]. In addition, strong insecticidal effects have been reported in fumigants and essential oils made from *C. incana* against the rice grain storage product pest, the common weevil, based on volatile component analysis using GC/MS [4]. Thus, in addition to its horticultural ornamental value, *C. incana* also has value as a functional plant.

*C. incana* is considered to be a type of continental plant that survives in Japan on the islands of Western Kyushu and the Western part of mainland Kyushu, and the native environment of *C. incana* in Japan is restricted to Western Kyushu [5,6]. *C. incana* is mainly native to positive open rocky areas and shows the characteristics of a rock crevice plant that grows in soil and rock crevices with accumulated organic matter and is sometimes observed in the same community as the Japanese spruce *Selaginella tamariscina* (P. Beauv.) Spring [6]. Given this native environment, each native population of *C. incana* is not contiguous but rather locally distributed and may exist in multiple local populations within the same area. In 2000, *C. incana* was listed in the Red Data Book of Japan as “Vulnerable II (VU)”, where “Vulnerable (VU)” follows the IUCN Red List category, and “II” refers to the national classification indicating the species is considered Vulnerable under Category II of Japan’s domestic conservation system [7]. Because the native population is declining due to the destruction of the native environment, caused mainly by land development, road construction, and trampling, etc., a survey of the native environment of *C. incana* in Japan was reported in 1988 [6]. Since there have not been any subsequent surveys on the distribution of *C. incana* since this survey, we conducted a survey on its native environment in Western Kyushu, Japan [8,9]. The survey revealed a total of 109 populations: 72 on Tsushima Island, Nagasaki Prefecture; 16 on the Nagasaki mainland; 16 on the Goto Islands; and populations in open rocky areas and other native environments on the Koshikijima Islands, Kagoshima Prefecture (Figure 2). In these areas, we observed a declining trend in native populations on the mainland of Nagasaki. In addition, a phylogenetic analysis of chloroplast DNA was used to classify haplotypes across native populations of *C. incana* in Western Kyushu. On the other hand, the phenotypic variation in this species in Western Kyushu has not been studied to date.

Previous studies on *C. incana* have primarily focused on its distribution and phylogenetic relationships [8,9]. However, comprehensive evaluations of phenotypic variation across its native range remain limited. *C. incana* shows the characteristics of a rock crevice plant, which means that the set of individuals recognizable as a population is localized, and growth is naturally in a spatially isolated state. Therefore, different phenotypic variations adapted to the native environment may have arisen in each region. The objective of this study was to investigate phenotypic variations in each native population through growth surveys and to evaluate phenotypic characteristics in each region and their relationship to the geographic structure. This enables a comprehensive understanding of the phenotypic diversity among native populations of *C. incana* in western Kyushu. Furthermore, elucidating the phenotypic diversity and adaptive strategies of *C. incana*, a species with a locally restricted distribution, may also contribute to understanding adaptive mechanisms in other plant species with similar ecological characteristics. The information obtained on phenotypic variation is expected to be useful not only as a genetic resource for breeding materials, but also for developing conservation plans for this endangered species.

## 2. Results

### 2.1. Comparison by Trait Classification

#### 2.1.1. Phenology

The mean number of days to flowering in all individuals was 168.62 ± 0.50 days (hereinafter, “±” indicates the standard error). The mean number of days to flowering by region was 165.12 ± 0.87 days for the Tsushima Islands populations (referred to as “TI”), 166.78 ± 0.76 days for the Nagasaki Mainlands populations (referred to as “NM”), 171.73 ± 0.87 days for the Goto Islands populations (referred to as “GI”), and 180.28 ± 1.75 days for the Koshikijima Islands populations (referred to as “KI”). The KI tended to bloom later than other regions (Table 1). In the number of days in the flowering period, no significant differences were observed among the regions.

#### 2.1.2. Florets and Flower Cluster Traits

The means and significance test results for the trait categories for individual flower forms and clusters by region are shown in Table 1. In addition, the results of the principal component analysis for the same trait categories are shown in Table 3. The results of principal component analyses showed that PC1 tended to have high principal component loadings for corolla lengths and vertical and horizontal diameters of flower clusters, suggesting that PC1 is an indicator of corolla and flower cluster size. On the other hand, PC2 tended to have higher principal component loadings for the L and b values of flower color and lower loadings for the value of flower color, suggesting that PC2 is an index related to the color shade of flowers. From the scatter plots of PC1 and PC2 scores (Figure 3), PC2 tended to be lower for TI, PC1 was lower, and PC2 was higher for NM and GI, and both PC1 and PC2 were higher for KI. The results of the significance difference test showed that the traits, except for male length and number of steps, significantly differed across regions. Comparing the size of individual corolla (corolla length, corolla width) and flower clusters (vertical and horizontal diameters of flower clusters), KI tended to be significantly larger, and NM and GI tended to be smaller in terms of flower form. Furthermore, comparing the number of florets per flower cluster, NM and GI showed less, KI showed more, and TI showed an intermediate number. In terms of flower color, NM and GI had bright, reddish-purple flowers due to large L and b values and small a values, whereas TI tended to have dark, bluish-purple flowers. The results of the above analysis showed that the characteristics of florets and flower cluster traits could be characterized by region, with TI having darker flowers, NM and GI having smaller flower parts and paler flower parts, and KI having larger flower parts.

#### 2.1.3. Leaf Traits

The means and significance test results for the trait categories for leaves by region are shown in Table 1. The results of the principal component analysis are shown in Table 4. The results of principal component analyses indicated that PC1 tended to have higher principal component loadings for leaf color or L and b values, and lower loadings for a values of leaf color, suggesting that it is an indicator related to leaf color shade. On the other hand, PC2 tended to have higher principal component loadings for leaf width, aspect ratios of leaves, and petiole width, suggesting that it is an indicator of leaf size in the lateral direction. From the scatter plots of PC1 and PC2 (Figure 4), PC1 tended to be slightly lower for TI, PC1 tended to be slightly higher for NM and GI, and PC2 tended to be higher for KI. The results of the significance difference test showed that the traits, except petiole length, significantly differed across regions. A comparison of leaf size showed that the leaves of NM tended to be smaller than those in other regions. KI showed smaller aspect ratios of leaves and larger leaf apical angles, indicating that they had broadly shaped leaves. The results of the above evaluations showed that leaf traits were characterized by region: darker leaves for TI, slightly smaller leaves for NM and GI (especially noticeably smaller for NM), brighter leaves for NM, and larger leaf width for KI.

Herbaceous form traits 

The means and significance test results for the trait categories related to herbaceous form by region are shown in Table 1. The results of the principal component analysis for the same trait categories are shown in Table 5. The results of principal component analysis showed that PC1 tended to have high principal component loadings for length between flower clusters, distance from the ground to the first flower cluster, plant height at flowering and fruiting, internode length, plant height at fruiting/number of flower clusters, and spike length/number of flower clusters on main stem-1, suggesting that it is an indicator of plant size in the vertical direction. On the other hand, PC2 tended to have higher principal component loadings for spike length/distance from the ground to the first flower cluster, and lower values for the ratio of height of plant at flowering/at fruiting and number of lateral flowering branches, suggesting that it is an indicator related to inflorescence development. From the scatter plots of PC1 and PC2 (Figure 5), PC1 tended to be higher for TI, PC1 was slightly lower for NM and GI, PC2 was lower, and KI was intermediate between PC1 and PC2 in the other regions. The results of the significance difference test showed that all the traits related to herbaceous form were significantly different across regions. Similarly to PC1 in the principal component analysis, NM and GI showed small values for traits related to longitudinal size, and TI showed large values. On the other hand, KI showed a significantly larger value for position of the first inflorescence node and a smaller value for the number of lateral flowering branches, similar to TI, indicating a unique herbaceous form with flower clusters concentrated at the top of the main stem. The values of length between flower clusters, internode length, and plant width significantly differed across all regions. The results of the above evaluations showed that TI had a large plant height and a few long inflorescences, NM and GI had a small plant height but grew horizontally and had many short inflorescences, and KI showed a unique characteristic in which the flower clusters were concentrated at the top of the main stem.

### 2.2. Phenotypic Variation Across and Within Regions

The coefficients of variation for each trait by region and for the entire population are shown in Table 2. The highest coefficient of variation in the entire population was the number of lateral flowering branches (0.46), followed by spike length/distance from the ground to the first flower cluster (0.44), number of days in the flowering period (0.42), number of florets (0.35) and spike length (0.32). The least important trait was the L-value of leaf color (0.07), followed by the number of days to flowering (0.07), the aspect ratio of the flower cluster (0.09), stamen length (0.10), and sepal length (0.11). The mean coefficient of variation for all traits in the entire population was 0.27. The traits that had coefficients of variation lower than the coefficients of variation in the entire population within all regions were peduncle width, number of florets, L value of flower color, leaf width, angle between the stem and petiole, plant height at fruiting, and spike length/number of flower clusters on main stem-1.

## 3. Discussion

### 3.1. Traits Related to Phenotypic Variation

In this study, to eliminate the influence of environmental variability and clarify genetic differences among populations, all individuals from different regions were cultivated and evaluated under uniform conditions. Because each regional population has distinct genetic and ecological characteristics, it was not appropriate to designate any one region as a universal control. This approach enabled a clearer comparison of region-specific phenotypic traits, primarily reflecting genetic divergence rather than environmental plasticity. The results of the growth survey in this study indicate that the native population of *C. incana* in Western Kyushu has diverse phenotypes. Principal component analyses by trait category identified several traits with high factor loadings that discriminated populations by region, and many of these traits showed significant differences between regions. The traits characterized by variations in the coefficient of variation were summarized below by trait category as traits related to phenotypic variations in this species.

The following section discusses key traits that exhibited notable variation across regions, based on the results presented in Section 2. Rather than reiterating the data, we aim to interpret these findings in the context of potential genetic differentiation and ecological adaptation. First, among the traits related to phenology, the coefficient of variation (0.06–0.08) of days to flowering within each region was smaller than that of other traits, indicating that the variation within regions was small and that the traits were common across populations in each region (Table 1 and Table 2). On the other hand, the coefficient of variation in number of days in flowering periods was relatively high in the entire population (0.43) and within each region (0.39–0.45), and there were no significant differences across regions, suggesting that this trait has large inter-individual and inter-population differences. Next, among the traits related to floral parts, the coefficient of variation for stamen length was relatively small in the entire population (0.10) and within each region (0.09–0.11), but there were no significant differences across regions, suggesting that the phenotypic variations in this species in Western Kyushu are small. On the other hand, the coefficient of variation (0.08–0.10) for L values of flower color, which differed significantly across regions, was lower within each region than the coefficient of variation (0.12) for the entire population. The coefficient of variation in the number of florets, which showed a high factor loading (0.72) in PC3 and was significantly different across regions, showed a relatively large value (0.35) in the entire population; however, the same characteristics as above were observed. These traits were inferred to be common traits across the populations within each region. These characteristics were also observed in L values of leaf color and leaf width among the leaf-related traits, suggesting that these traits were common across the populations in each region. Among the traits related to herbaceous form, the coefficient of variation for the number of lateral flowering branches in the entire population was highest (0.46) among all traits. When this coefficient of variation was compared across regions, the values were relatively high for TI (0.47) and KI (0.48) and relatively low for NM (0.38) and GI (0.34), suggesting that the feature of significantly higher number of lateral flowering branches was a common phenotype to NM and GI. On the other hand, the number of florets (per single flower cluster) was smaller for NM and GI than in other regions, suggesting that the total number of florets per individual was similar to that in other regions.

### 3.2. Summary of Specific Phenotypes in Each Region

Based on the aforementioned references to each trait, the phenotypes in each region are summarized again. The results of principal component scores in each population by trait category by principal component analysis indicated specific phenotypes in each region (Figure 3, Figure 4 and Figure 5, Table 3, Table 4 and Table 5).

First, the TI phenotype was characterized by a long, narrow herbaceous form with a few long inflorescences. In addition, flowering was early, and both flowers and leaves were dark in color. Next, NM and GI were characterized by their common phenotype of having a horizontally spreading herbaceous form with numerous short inflorescences that grow laterally instead of being small in height. In addition, both flowers and leaves tended to be small and pale in color. On the other hand, the different phenotypes of NM and GI showed no differences in plant height; however, NM had shorter internodes and a smaller plant width, resulting in a smaller herbaceous form. Moreover, NM tended to flower a little earlier and have a narrower corolla, shorter pistil, and smaller leaves. Finally, the KI phenotype was characterized by inflorescence with a small number of concentrated upper flower clusters, which differed significantly from other regions. In addition, both flowers and leaves were intermediate in color but showed characteristics of larger corollas and clusters, as well as larger leaf widths.

### 3.3. Factors in the Formation of Phenotypic Variation

The relationship between the native environment and the timing of distribution was discussed as a factor in the formation of phenotype of this species in each region of Western Kyushu. *C. incana* is mainly native to dry, sunny, exposed rocky regions where other plants cannot grow because it exhibits characteristics similar to those of crevice plants. Given this native environment, each native population of this species was not contiguous but rather locally distributed and may have been divided into several local populations within the same region. However, the main native environment in Tsushima consisted of positive environments such as roadsides and natural gaps, and the high water content was thought to cause competition with other plants for survival. Furthermore, the high population density in the area may have resulted in genetic hybridization across the populations (Table 6). This native environment was thought to have given TI a long and slender herbaceous form with long inflorescences to compete with other plants for sunlight and to attract pollinators as a phenotype of TI. On the other hand, the main native environment on the Nagasaki mainland and the Goto Islands are dry and sunny environments, such as near mountain tops and on exposed rocks along the coastline, where there are few other plants, and competition was inferred to be low. Furthermore, the population density within the region was low, suggesting that genetic exchange occurred only within each population (Table 6). This native environment was thought to have led to the small herbaceous form phenotype of NM and GI since these did not need to compete for survival with other plants and could conserve resources in a harsh environment. In particular, NM were found in many native areas at high elevations, suggesting that these acquired a smaller herbaceous form in order to adapt to strong winds (Table 6).

The main native environment in the Koshikijima Islands was not unique compared to other areas. On the other hand, the KI phenotype was distinctive and differed significantly from that of other regions. In particular, KI showed significantly smaller values than TI in terms of plant height at flowering, while KI showed similarly large values as TI in terms of plant height at fruiting, with the highest tendency being observed at the position of the first inflorescence node, indicating that KI rapidly changes the plant height after flowering. KI showed the highest value in the number of days to flowering, indicating that these bloomed later and more intensively than species in other regions. This characteristic was probably acquired to adapt to the native environment of KI that was often affected by typhoons in the summer season, leading to flowering being delayed compared to other regions to reduce the risk of flowers being lost immediately after flowering, while at the same time ensuring an adequate seed maturation period. In addition, the KI populations exhibited both significantly larger floral organs (e.g., corolla and flower cluster dimensions) and higher L and b values in flower color, indicating brighter and more vivid coloration. This combination of large and bright flowers may enhance pollinator attraction, particularly in the KI region where flowering is delayed and concentrated, possibly as an adaptation to seasonal climatic disturbances such as typhoons.

Traits related to reproductive fitness, such as inflorescence, tended to have high coefficients of variation and were inferred to be relatively variable in concert with the geographic variation for phenotypes of this species. On the other hand, significant differences were also observed across regions for traits related to flower and leaf color, which are relatively less variable and have small coefficients of variation within and across regions. Phenotypic variation for these traits may be due to geographical variation caused by environmental factors, as well as the maintenance of ancestral lineage characteristics when the distribution spread across various regions. In addition, a chloroplast haplotype analysis suggested that the ancestral lineage of KI may be different from other domestic lineages, suggesting that it has a unique current phenotype [8,9]. Recent studies have highlighted the importance of phenotypic diversity in plant adaptation to environmental gradients, including aridity and temperature variation (e.g., ref. [10]). These findings support the relevance of evaluating intraspecific variation in endangered species such as *C. incana*.

### 3.4. Usefulness as a Plant Resource and Conservation

Native populations of *C. incana* in Western Kyushu showed different phenotypic variations across regions. The various phenotypes in each region were evaluated with the expectation that these plant resources would be used as breeding material. The horticultural varieties of *C. incana* have limited flower colors: blue-purple, white, and peach, which have been confirmed in their native environment [8,9]. On the other hand, although there are some variations in height across these varieties, few varieties have been produced that are distinguished by inflorescence, form, and leaf color. The phenotypic variability of this species in Western Kyushu was inferred to be useful as breeding material for the production of horticultural varieties. First, TI, with its tall, vertical, slender grass form, was assumed to be suitable for breeding varieties for border gardens that provide color after the summer, when flowers are scarce. Next, NM and GI, which have smaller herbaceous forms, may be proposed as suitable phenotypes for compact potted varieties because they show a high number of lateral flowering branches and many inflorescences with small flower clusters. KI, with its large flower clusters and tendency to flower intensively in a short period of time, could be used as material for the creation of cut flower varieties. In addition, the color shades of the flowers and leaves were also suggested to be effective as breeding materials. These variations are assumed to be useful not only for breeding horticultural varieties of this species but also for breeding Caryopteris hybrids with this species as a single parent. In addition, phenotypic diversity has been reported to contribute to resilience against environmental disturbance in endangered species [11], suggesting that the regional and intraspecific variation observed in this study may also inform conservation strategies. On the other hand, the native population of this species is declining due to the destruction of its native environment, and it is classified as “Vulnerable II (VU)” in Japan. The native environment surveys conducted by the authors also confirmed a decrease in the native population from the previous survey results [8,9]. Since plants’ responses to such anthropogenic environmental changes are delayed, the inference is that the potential for extinction is more advanced than the current native status [12,13]. Although eliminating all anthropogenic factors such as land development, road construction, trampling, etc., and conserving the native environment of all populations in Western Kyushu are not feasible, the establishment of conservation priorities will contribute to the planning of future conservation programs, including ex situ conservation. Furthermore, the observed phenotypic variation across and within regions suggests the presence of distinct genetic pools that may reflect local adaptations. Incorporating this phenotypic diversity into conservation planning is essential to preserve not only the species but also its adaptive potential. Populations exhibiting unique or extreme phenotypes—such as the KI populations with large and bright flowers—should be prioritized for conservation, as they may harbor traits valuable for both ecological resilience and future breeding efforts.

The results of this study showed a variety of phenotypes in each region by trait category. Furthermore, phenotypes with high coefficients of variation and wide ranges of variation were observed even within the same region, suggesting the existence of various gene pools for different uses. Considering the phenotypic variation in each population in this study, as well as information on the autochthonous environment status, such as chloroplast haplotypes [8,9] and the number of individuals within a population, the establishment of conservation priorities for each native population in each region and area could contribute to maintaining the intraspecific diversity of *Caryopteris incana*. These findings are expected to provide a foundation for future conservation planning and for the selection of breeding materials in horticultural development. Furthermore, to better understand the species’ adaptive potential and genetic background in the context of climate change, future studies should incorporate molecular markers, genomic data, and integrated analyses with climatic variables.

## 4. Materials and Methods

### 4.1. Growth Survey

From the 72 populations identified in Tsushima, 20 populations were selected from across the entire area to avoid geographical bias. Additionally, 37 populations from various regions of Western Kyushu, excluding Tsushima, were selected, making a total of 57 populations used for the growth survey. The smaller number of populations from the Koshikijima Islands reflects the limited natural distribution of *C. incana* in that region. The methods for selecting populations and defining population units in each region were based on our previous studies [8,9], and are not repeated in detail here. Information on native environments in each region is shown in Table 6. After sowing, the plants were grown under natural sunlight in a greenhouse. First, 50 grains of each population were sown on filter paper with Gibberellin A3 in a Petri dish and maintained at 25 °C humidity. After approximately one to two weeks, the germinated plants were transplanted into cell trays and maintained in a shaded growing tunnel. About one month later, after confirming 4–6 true leaves, the plants were potted into No. 2.5 pots using a culture medium consisting of red ball soil, peat moss, and perlite at a ratio of 7:2:1. After confirming 8–12 true leaves, up to 12 individuals in each group were potted into No. 7 pots and maintained until measurement. Primary fertilizers were not used to maintain the local environment, but a liquid fertilizer, Hyponex (N-P-K = 6-10-5, Hyponex Co., Ltd., Tokyo, Japan), diluted 2000-fold, was added to Pot Nos. 2.5 and 7 every two weeks to once a month. One medium-sized fertilizer tablet (N-P-K = 12-12-12, Hyponex Japan) was added to each pot in August and October.

### 4.2. Measurement Method

About 6 months after sowing, measurement surveys were initiated as flowering began. Digital calipers and tape measures were used for each measurement. Thirty-six survey characteristics were selected. Flowering was considered to have occurred once the corolla lobes of the first flower were completely open, and the number of days from sowing was calculated using the number of days to flowering. Next, two flowers were randomly selected for each individual in each population, and seven vase morphology characteristics were evaluated. The survey characteristics were corolla length, corolla width, stamen length, pistil length, sepal length, peduncle length, and peduncle width. Corolla width was measured as the longitudinal distance from the top of the flower to the tip of the lip petal (Figure 6A). Stamens and pistils were measured by dissecting each flower. Eight leaf morphological parameters were evaluated on one leaf at the 10th internode of each individual that had fully grown to adulthood. The survey characteristics were leaf length, leaf width, petiole length, petiole width, number of serrations, depth of serration, angle of the leaf tip, and angle between the stem and petiole. The depth of serration was measured as the distance between the tip of the fifth serrated tooth from the leaf apex and the previous serrated tooth. The leaf tip angle was measured as the angle between the lines connecting the leaf tip and the fifth serration from the tip on each side of the leaf, and the stem and petiole angle was measured as the angle between the stem and petiole on the side of the axis of orientation (Figure 6B). Once all florets of the first flower cluster of each individual were in full bloom, nine herbaceous form parameters were evaluated at the time of flowering. The survey characteristics were number of florets, vertical diameter of flower clusters, horizontal diameter of flower clusters, length between flower clusters, distance from the ground to the first flower cluster, position of the first inflorescence node, plant height at flowering, plant width, and internode length. The number of florets, the vertical diameter of the flower cluster, and the horizontal diameter of the flower cluster were measured using the first flower cluster of the main stem. The distance between the first and second flower clusters was measured from the first flower clusters to one node down (Figure 6C). Once all flowers of each individual were completely closed, the number of days from flowering was calculated as the number of days in the flowering period, and nine herbaceous morphological parameters were evaluated at the time of flower closure. The survey characteristics were the number of flower clusters on the main stem, plant height at fruiting, number of lateral flowering branches, and stem diameter. Three flower color parameters and three leaf color parameters were investigated using a CM-700d spectrophotometer (KONICA MINOLTA, INC.). The survey characteristics were L and a, b values of flower color, and L and a, b values of leaf color using Hunter Lab values. Color traits of flowers and leaves were measured using the CIE Lab color space, which is widely used in color science. In this system, L represents lightness (ranging from 0 = black to 100 = white), a represents the green–red axis (negative values indicate green, positive values indicate red), and b represents the blue–yellow axis (negative values indicate blue, positive values indicate yellow). Leaves were measured at the time of leaf morphometry, and flowers were measured from the lip petals of fresh flowers up to one day after flowering. Seven characteristics related to flower form, such as corolla length; thirteen characteristics related to plant shape, such as plant height and plant width; and ten individuals in each population were evaluated for days to flowering and days in the flowering period. In addition, eight characteristics related to leaves, such as leaf height, and six characteristics related to color, such as flower color, were measured with a target of five individuals in each population. The 10th internode was selected to ensure that the leaf had reached full maturity, minimizing developmental variation among individuals. Similarly, the 5th serration from the leaf apex was used as a consistent reference point to standardize measurements of serration depth and leaf tip angle across samples. Nine additional secondary variables were added to these characteristics, for a total of 45 characteristics compared in each group. To summarize the sampling design, a total of 20 populations from TI, 16 from NM, 16 from GI, and 5 from KI were surveyed. For each population, the number of individuals evaluated per trait category was as follows: 10 individuals for phenology traits, 10 individuals for floral traits (including 5 individuals for flower color measurements), 5 individuals for leaf traits, and 10 individuals for herbaceous form traits. These sampling numbers were determined to ensure consistency and comparability across populations and trait categories.

### 4.3. Statistical Analysis

A principal component analysis using IBM SPSS Statistics 30.0^®^ (IBM Analytics) was performed to compare the populations. A one-way analysis of variance by trait category was performed to determine significant differences in traits across regions. If equal variances were assumed, Tukey B was used, Hochberg’s GT2 was used if the sample sizes were very different, and Dunnet T3 was used if equal variances were not assumed.

## Figures and Tables

**Figure 1 plants-14-01840-f001:**
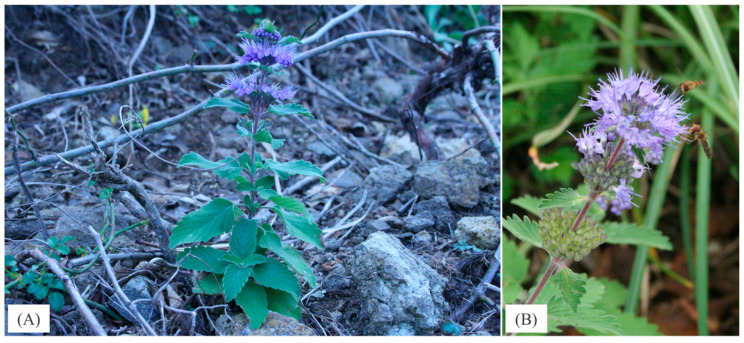
*Caryopteris incana* (Thunb. ex Houtt.) Miq. in its native habitat (**A**) Whole plant: 30 cm to over 1 m tall, showing both erect and stoloniferous growth (**B**) Close-up of flower: fragrant blue-purple flowers borne in axillary inflorescences.

**Figure 2 plants-14-01840-f002:**
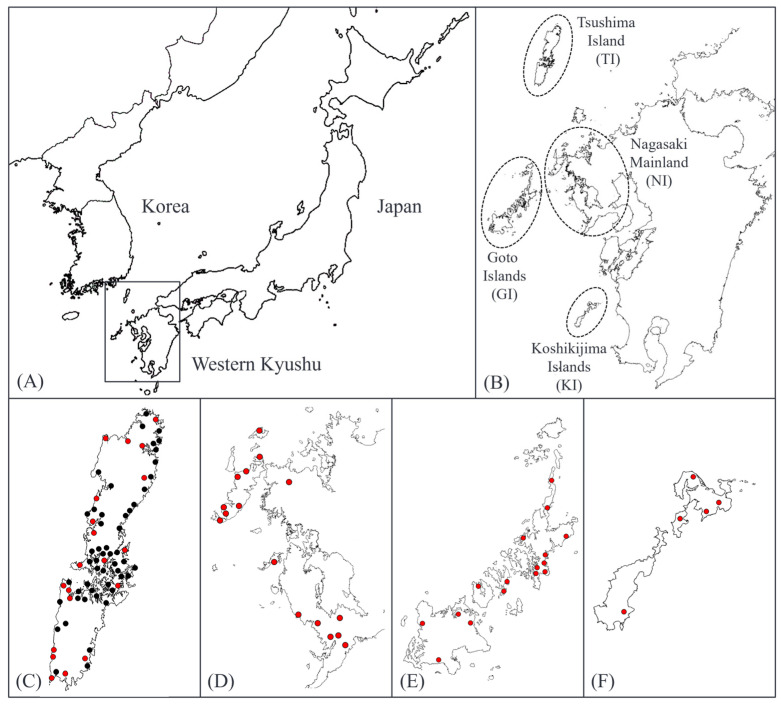
Geographic distribution of *C. incana* populations in Western Kyushu, Japan. (**A**) Overview map showing the location of Western Kyushu within Japan. (**B**) Detailed map indicating the names of regions (Tsushima Island, Nagasaki Mainland, Goto Islands, Koshikijima Islands) and the locations of native populations surveyed in this study. (**C**–**F**) Regional maps showing the locations of all known native populations in each area: (**C**) Tsushima Islands (TI), (**D**) Nagasaki Mainland (NM), (**E**) Goto Is-lands (GI), and (**F**) Koshikijima Islands (KI). In the Tsushima region, 72 populations have been recorded, of which, 20 populations used in this study are indicated in red.

**Figure 3 plants-14-01840-f003:**
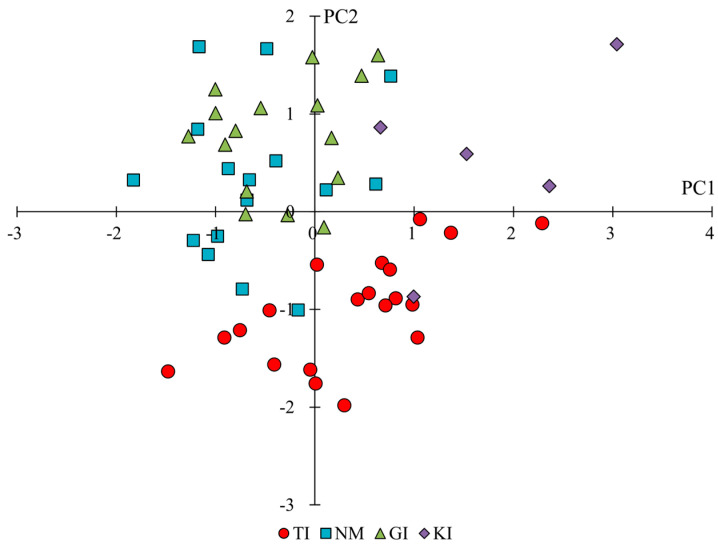
Scatterplot of principal component scores for the traits of florets and flower clusters of each population. PC1 represents corolla and flower cluster size, while PC2 reflects flower color characteristics (L, a, b values).

**Figure 4 plants-14-01840-f004:**
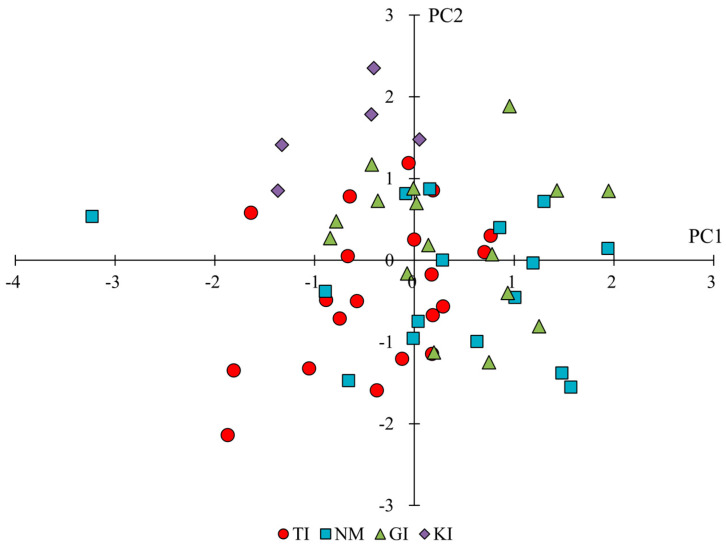
Scatterplot of principal component scores for leaf traits of each population. PC1 reflects leaf color (L, a, b values), and PC2 represents leaf width and shape.

**Figure 5 plants-14-01840-f005:**
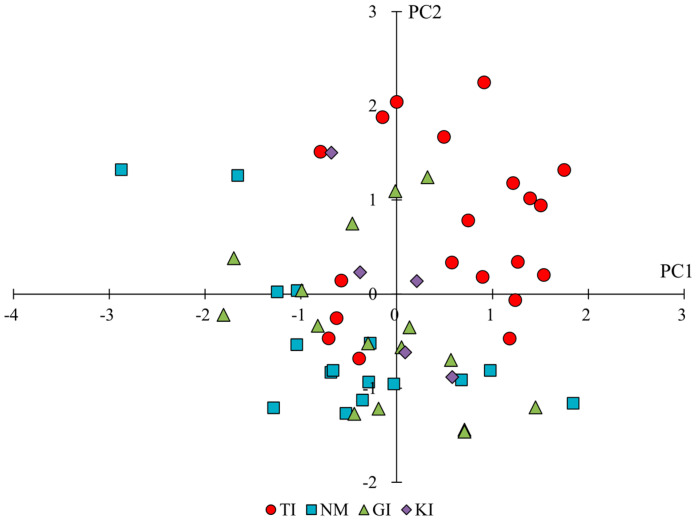
Scatterplot of principal component scores for the trait of herbaceous form of each population. PC1 indicates vertical plant size, and PC2 reflects inflorescence development.

**Figure 6 plants-14-01840-f006:**
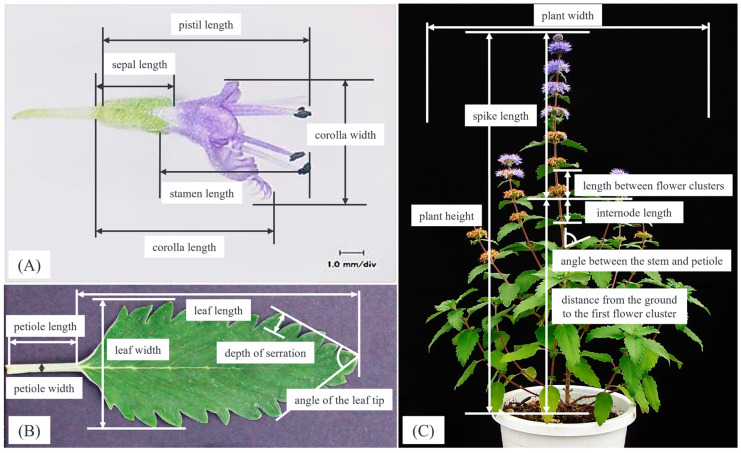
Measured parts (**A**) Flowers (**B**) Leaves (**C**) Entire plant.

**Table 1 plants-14-01840-t001:** Means and variance analysis results by trait category across regions in the growth survey. Superscript letters (a, b, c) indicate statistically significant differences among regions (*p* < 0.05). Standard errors are not shown; for trait variability, please refer to Table 2.

A	mean value of survey traits for phenology																													
group	A1: number of days to flowering [d]					A2: number of days in the flowering period [d]																								
Tsushima Islands (TI)	165.12	a				77.19	a																							
Nagasaki Mainland (NM)	166.78	a				88.87	a																							
Goto Islands (GI)	171.73		b			83.5	a																							
Koshikijima Islands (KI)	180.28			c		76.23	a																							
All population	168.62					81.83																								
B	mean value of survey traits for florets and flower cluster																													
group	B1: corolla length [mm]					B2: corolla width [mm]					B3: aspect ratio of corolla (B1/B2)					B4: stamen length [mm]					B5: pistil length [mm]					B6: ratio of stamen/pistil (B4/B5)				
TI	9.51		b			7.35	a				1.31			c		7.54	a				10.75		b			0.71		b		
NM	9.01	a				7.43	a				1.23		b			7.55	a				10.29	a				0.74		b		
GI	8.99	a				7.92		b			1.15	a				7.75	a				10.93		b			0.72		b		
KI	10.26			c		8.59			c		1.21	a	b			7.71	a				11.87			c		0.65	a			
All population	9.28					7.63					1.24					7.61					10.75					0.72				
B	mean value of survey traits for florets and flower cluster																													
group	B8: peduncle length [mm]					B9: peduncle width [mm]					B10: number of florets					B11: number of flower clusters on main stem					B12: vertical diameter of flower clusters [mm]					B13: horizontal diameter of flower clusters [mm]				
TI	8.46	a	b			1.48	a				73.09		b			8.52	a				44.35		b			38.98	a			
NM	8.1	a				1.49	a				66.73	a				8.72	a				41.66	a				37.58	a			
GI	8.77		b	c		1.44	a				61.46	a				8.14	a				42.48	a	b			38.9	a			
KI	9.26			c		1.74		b			87.17			c		9.04	a				50.62			c		43.6		b		
All population	8.51					1.49					69.09					8.51					43.53					38.91				
B	mean value of survey traits for florets and flower cluster																													
group	B15: flower color L-value					B16: flower color a-value					B17: flower color b-value																			
TI	49.29	a				10.46		b			−23.05			c																
NM	56.72			c		7.6	a				−17.36	a																		
GI	58.2			c		7.53	a				−17.93	a	b																	
KI	53.81		b			9.8		b			−19.94		b																	
All population	54.32					8.76					−19.71																			
C	mean value of survey traits for leaf																													
group	C1: leaf length [mm]					C2: leaf width [mm]					C3: aspect ratio of leaf (C1/C2)					C4: petiole length [mm]					C5: petiole width [mm]					C6: number of serrations				
TI	65.85		b			33.26	a	b			2			c		19.11	a				2.2	a				19.54			c	
NM	61.12	a				32.34	a				1.93		b	c		18.86	a				2.2	a				16.57		b		
GI	64.39		b			34.8		b			1.87		b			19.79	a				2.31		b			15.82	a	b		
KI	63.48	a	b			41.23			c		1.55	a				20.06	a				2.36		b			14.64	a			
All population	63.89					34.06					1.91					19.3					2.24					17.28				
C	mean value of survey traits for leaf																													
group	C8: angle of the leaf tip [°]					C9: angle between stem and petiole [°]					C10: leaf color L-value					C11: leaf color a-value					C12: leaf color b-value									
TI	63.92		b			66.97	a				36.4	a				−7.62		b			15.55	a								
NM	59.92	a				77.95		b			38.25			c		−8.04	a				17.09		b							
GI	59.06	a				77.66		b			37.61		b	c		−8.21	a				17.25		b							
KI	67.19			c		74.7		b			37.1	a	b			−7.8	a	b			14.83	a								
All population	61.72					73.75					37.32					−7.91					16.4									
D	mean value of survey traits for herbaceous form																													
group	D1: length between flower clusters [mm]					D2: distance from ground to the first flower cluster mm]					D3: spike length (D6-D2)					D4: position of first inflorescence node					D5: plant height at flowering [mm]					D6: plant height at fruiting [mm]				
TI	41.32			c		258.04		b			282.41		b			15.66	a				441.16		b			540.45		b		
NM	29.77	a				229.19	a				221.65	a				16.32		b			364.75	a				450.91	a			
GI	34.37		b			229.73	a				230.27	a				16.47		b			369.42	a				460	a			
KI	29.16	a				271.65		b			251.93	a	b			18.67			c		390.57	a				523.59		b		
All population	35.07					242.8					247.81					16.31					394.98					490.64				
D	mean value of survey traits for herbaceous form																													
group	D8: plant width [mm]					D9: internode length [mm]					D10: number of flowering side branches					D11: stem diameter [mm]					D12: D6/B11					D13: D3/D2				
TI	290.66		b			38.71			c		5.34	a				8.16		b			66.03			c		1.17		b		
NM	319.95			c		28.98	a				9.04		b			8.06	a	b			54.32	a				1.08	a	b		
GI	352.95				d	32.52		b			8.3		b			7.67	a				59.69	a	b			1.08	a	b		
KI	243.43	a				28.85	a				5.11	a				7.55	a				60.75		b	c		0.98	a			
All population	312.88					33.38					7.22					7.95					60.44					1.1				

**Table 2 plants-14-01840-t002:** Coefficient of variation for each trait. Labels for each trait indicate the same labels as in Table 1.

A	coefficient of variation traits for phenology													
group	A1	A2												
Tsushima Islands (TI)	0.08	0.44												
Nagasaki Mainland (NM)	0.06	0.39												
Goto Islands (GI)	0.06	0.45												
Koshikijima Islands (KI)	0.07	0.41												
All population	0.07	0.43												
B	coefficient of variation for traits for florets and flower cluster													
group	×1	×2	×3	×4	×5	×6	×7	×8	×9	×10	×11	×12	×13	×14
TI	0.09	0.14	0.15	0.1	0.15	0.12	0.11	0.25	0.12	0.34	0.26	0.14	0.14	0.08
NM	0.11	0.13	0.14	0.1	0.13	0.12	0.09	0.29	0.13	0.33	0.25	0.14	0.11	0.09
GI	0.11	0.13	0.16	0.11	0.12	0.14	0.12	0.23	0.11	0.32	0.31	0.13	0.12	0.11
KI	0.09	0.1	0.14	0.09	0.1	0.11	0.08	0.16	0.12	0.32	0.28	0.15	0.11	0.09
All population	0.11	0.14	0.16	0.1	0.14	0.13	0.11	0.25	0.13	0.35	0.27	0.15	0.13	0.09
B	coefficient of variation for traits for florets and flower cluster													
group	×39	×40	×41											
TI	0.09	0.19	0.12											
NM	0.1	0.27	0.3											
GI	0.09	0.23	0.25											
KI	0.08	0.15	0.16											
All population	0.12	0.27	0.24											
C	coefficient of variation for traits for leaf													
group	×27	×28	×29	×30	×31	×32	×33	×34	×35	×36	×37	×38		
TI	0.11	0.13	0.12	0.23	0.12	0.14	0.3	0.13	0.14	0.07	0.16	0.21		
NM	0.1	0.15	0.17	0.19	0.1	0.21	0.32	0.14	0.15	0.07	0.15	0.2		
GI	0.13	0.13	0.15	0.22	0.11	0.21	0.29	0.12	0.16	0.06	0.11	0.17		
KI	0.11	0.11	0.11	0.19	0.12	0.15	0.36	0.11	0.15	0.06	0.11	0.17		
All population	0.12	0.15	0.16	0.21	0.12	0.2	0.31	0.14	0.16	0.07	0.14	0.2		
D	coefficient of variation for traits for herbaceous form													
group	×15	×16	×17	×18	×19	×20	×21	×22	×23	×24	×42	×43	×44	×45
TI	0.24	0.25	0.29	0.14	0.21	0.19	0.15	0.38	0.27	0.47	0.17	0.23	0.41	0.2
NM	0.29	0.31	0.27	0.14	0.27	0.2	0.16	0.23	0.34	0.38	0.21	0.29	0.49	0.19
GI	0.23	0.27	0.33	0.16	0.21	0.21	0.15	0.27	0.27	0.34	0.19	0.26	0.43	0.17
KI	0.18	0.19	0.34	0.11	0.17	0.18	0.18	0.31	0.16	0.48	0.17	0.22	0.46	0.14
All population	0.29	0.27	0.32	0.15	0.24	0.21	0.15	0.32	0.31	0.46	0.19	0.26	0.44	0.22

**Table 3 plants-14-01840-t003:** Eigenvalues, contributions, and principal component loadings of each principal component for traits of florets and flower clusters.

Label	Trait	Characteristic value		
	principal component numbers	PC 1	PC 2	PC 3
	Eigenvalue	5.39	2.71	1.99
	Contribution rate (%)	31.70	15.96	11.69
B1	corolla length	0.72	−0.24	−0.28
B2	corolla width	0.64	0.52	−0.35
B3	aspect ratio of corolla (B1/B2)	−0.11	−0.71	0.17
B4	stamen length	0.39	0.34	−0.35
B5	pistil length	0.78	0.10	−0.49
B6	ratio of stamen/pistil (B4/B5)	−0.61	0.15	0.31
B7	sepal length	0.53	−0.16	−0.34
B8	peduncle length	0.28	0.37	0.29
B9	peduncle width	0.50	0.15	0.18
B10	number of florets	0.49	0.10	0.72
B11	number of flower clusters on the main stem	−0.23	−0.15	−0.10
B12	vertical diameter of flower cluster	0.84	0.30	0.40
B13	horizontal diameter of flower cluster	0.80	0.28	0.12
B14	aspect ratio of flower cluster (B12/B13)	0.39	0.13	0.59
B15	flower color (L-value)	−0.45	0.70	−0.06
B16	flower color (a-value)	0.66	−0.62	0.11
B17	flower color (b-value)	−0.52	0.67	−0.07

**Table 4 plants-14-01840-t004:** Eigenvalues, contributions, and principal component loadings of each principal component for traits of leaves.

Label	Trait	Characteristic value		
	principal component numbers	PC 1	PC 2	PC 3
	Eigenvalue	2.93	2.60	2.18
	Contribution rate (%)	24.40	21.64	18.17
C1	leaf length	0.36	0.20	0.79
C2	leaf width	−0.14	0.92	0.06
C3	aspect ratio of leaf (C1/C2)	0.36	−0.80	0.40
C4	petiole length	0.26	0.19	0.53
C5	petiole width	0.05	0.80	0.22
C6	number of serrations	0.14	−0.29	0.66
C7	depth of serration	0.32	0.31	0.25
C8	angle of the leaf tip	−0.26	0.39	0.15
C9	angle between the stem and petiole	0.29	0.08	−0.64
C10	leaf color (L-value)	0.86	0.05	−0.21
C11	leaf color (a-value)	−0.84	−0.21	0.22
C12	leaf color (b-value)	0.93	0.05	−0.17

**Table 5 plants-14-01840-t005:** Eigenvalues, contribution ratio, and principal component loadings of each principal component for the trait of herbaceous form.

Label	Trait	Characteristic value		
	principal component numbers	PC 1	PC 2	PC 3
	Eigenvalue	6.14	2.82	1.36
	Contribution rate (%)	43.88	20.18	9.68
D1	length between flower clusters	0.89	0.27	−0.07
D2	distance from the ground to the first flower cluster	0.90	−0.22	0.12
D3	spike length (D6-D2)	0.37	0.81	0.37
D4	position of the first inflorescence node	0.13	−0.38	0.07
D5	plant height at flowering	0.91	−0.08	0.10
D6	plant height at fruiting	0.82	0.36	0.31
D7	ratio of plant height at flowering/at fruiting (D5/D6)	0.47	−0.6	−0.29
D8	plant width	0.25	−0.38	0.75
D9	internode length	0.89	0.20	−0.04
D10	number of flowering side branches	−0.12	−0.67	0.54
D11	stem diameter	−0.28	0.11	0.00
D12	plant height at fruiting/number of flower clusters (D6/B11)	0.91	−0.21	−0.21
D13	spike length/distance from the ground to the first flower cluster (D3/D2)	−0.51	0.77	0.21
D14	spike length/number of flower clusters on the main stem-1 (D3/B11-1)	0.80	0.29	−0.23

**Table 6 plants-14-01840-t006:** Information on the native environments of the 57 populations in Western Kyushu.

Region	Altitude (m)				Number of Neighbor Populations			Number of Individuals		
	Ave.	Number of Populations								
		<10	10–100	100<	0	1	2≦	<20	20–100	100<
Tsushima Islands	39.9 ± 9.6	5	13	2	1	2	17	7	3	10
Nagasaki Mainland	175.3 ± 26.9	3	1	12	9	5	2	4	6	6
Goto Islands	35.8 ± 7.7	5	11	0	9	3	4	6	3	7
Koshikijima Islands	123.8 ± 22.0	0	3	2	3	2	0	3	0	2
whole region	84.1 ± 11.9	13	28	16	22	12	23	20	12	25

## Data Availability

The datasets generated and/or analyzed during the current study are available from the corresponding author on reasonable request. The data are not publicly available due to the enormous volume of measurement data for each item.

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
