# Peer review of "Intraspecific Evaluation of Phenotypic Variations of Caryopteris incana (Thunb. ex Houtt.) Miq. in Western Kyushu, Japan"

_plants, 2025, doi:10.3390/plants14121840_

Round 1

Reviewer 1 Report (Previous Reviewer 1)

Comments and Suggestions for Authors

plants-3630099- titled “ Intraspecific evaluation of phenotypic variations of Caryopteris incana in Western Kyushu, Japan

My comments on the manuscript are as follow:

  1. I have been through the manuscript thoroughly, the paper is important and may of interest to a wider community and worth reporting. The study is focusing on an endangered species Caryopteris incana native to parts of Western Kyushu Japan. Reporting of similar studies are of immense significance particularly when global climate concerns and biodiversity loss are at the top of UN SDGs/agenda. My suggestions are as under:

  1. Title: The title is OK , valid authority may be added along the scientific name.

  1. Abstract: Is clear and communicates well. However, there is no information of the current population size, only previous study is with 109 populations is stressed? There must be a brief (but essence) of the methodology undertaken and the rationale of the study with explicit objectives in the abstract. Additionally, there must a way forward at the end, what are the implications of this study? This may be added as the concluding remarks in abstract section as well as at the end of the MS. These recommendations may be well integrated into futuristic studies/guidelines for following researches; so that other studies/researchers may link their studies to the one here, and may get benefits of this study.

  1. Introduction:

  • Introduction section is written well with appropriate references. The study and subsequent findings can inform conservation strategies by identifying region-specific traits and genetic diversity, aiding in targeted conservation efforts as well as could add to breeding potential of the species.
  • Yet some of the highly related literature to explain/highlight the gaps are missing/non-existing and such references may be added in the MS.
  • The authors have explicitly mentioned the objectives at the end but why achieving these objectives are important? is missing and may be overseen.
  • Figure/s overall have good resolution but poorly described (see Fig. 1) and that needs attention to make figures/tables (here as well as elsewhere) self-explanatory.
  •  

  1. Results and Discussion:
  • The details are appropriate and sufficient, supported by related figures and tables. Still, is there any control possible for such experiments? If yes how control were selected, and if not why? More importantly how appropriate could the results be in any of the scenarios?
  • One of my concerns is, to provide rationale of why Caryopteris incana plants of different areas are grown and studied at one place when different morphotypes are already known/defined for the same species and the role of geography in inducing plastic variability is stressed by the authors themselves throughout the MS? This may be explained in any appropriate section of the MS.
  • It is worthwhile to see if the study design is appropriate to answer the research question (including the use of appropriate controls), and are the conclusions supported by the evidence presented in similar work before?
  • The authors have indicated that they has DNA based data (if I perceived it correctly?). While phenotypic traits are well-documented, the study could get benefits of such data by integrating genomic data to confirm whether observed variations are genetically fixed or environmentally induced and are plausible?.
  • While of speaking of the genetic/morphological variability and then running a test for significance on such data sets. One limitation is the sample size and regions (e.g., Koshikijima) had fewer populations, potentially skewing results. Is there any reason for this selection? This may be provided at any relevant section in the MS.
  • It is ascertained that greenhouse growth conditions may not fully replicate native environments required for Caryopteris incana, possibly masking some adaptive traits. Why not or why the experiment was taken in field-based trial?
  • The overall strength of the study is that, this study provides a baseline data/study for climatic resilience in Caryopteris incana. Still no climatic variables (e.g., precipitation, temperature) are correlated with phenotypic traits to predict responses to climate change. This may be overseen and if such data is available with the authors, appropriate details may be mentioned in the MS at any relevant section.

  1. Materials and Method sections is appropriate and judiciously placed with relevant references.
  2. There is no way forward/recommendations based on this study and may be added.

Decision:

While the study is within the scope of the journal, and information may be handy and of wider interest. The MS may be accepted for position after the suggested corrections/amendments.

Author Response

Comments 1: Title: The title is OK , valid authority may be added along the scientific name.

Response 1: The authority for the scientific name has been added to the title.

Comments 2: Abstract: Is clear and communicates well. However, there is no information of the current population size, only previous study is with 109 populations is stressed?

Response 2: The information on the 109 populations used in this study is based on field surveys and sampling conducted by the authors, as reported in our previous studies (References 8 and 9), which represent the most recent available data. Changes in habitat conditions and population decline since earlier surveys (References 5 and 6) have been discussed in those prior publications. The present study builds upon these findings to evaluate phenotypic variation.

Comments 3: There must be a brief (but essence) of the methodology undertaken and the rationale of the study with explicit objectives in the abstract. Additionally, there must a way forward at the end, what are the implications of this study?

Response 3: The abstract has been revised to clearly and concisely convey the research objectives, methodological highlights, significance of the study, and future perspectives.

Comments 4: The authors have explicitly mentioned the objectives at the end but why achieving these objectives are important? is missing and may be overseen.

Response 4: To clarify the importance of the research objective, we revised the final part of the Introduction to better explain the background and significance of the study.

Comments 5: Figure/s overall have good resolution but poorly described (see Fig. 1) and that needs attention to make figures/tables (here as well as elsewhere) self-explanatory.

Response 5: The captions for Figures 1 and 2 have been revised to improve clarity and ensure that the content is more self-explanatory.

Comments 6: Still, is there any control possible for such experiments? If yes how control were selected, and if not why?

Response 6: A rationale has been added at the beginning of the “3. Discussion” section.

Comments 7: One of my concerns is, to provide rationale of why Caryopteris incana plants of different areas are grown and studied at one place when different morphotypes are already known/defined for the same species and the role of geography in inducing plastic variability is stressed by the authors themselves throughout the MS?

Response 7: Additional explanations have been included in conjunction with the relevant section for Comment 6.

Comments 8: The authors have indicated that they has DNA based data (if I perceived it correctly?). While phenotypic traits are well-documented, the study could get benefits of such data by integrating genomic data to confirm whether observed variations are genetically fixed or environmentally induced and are plausible?.

Response : We agree with your comment that integrating genomic information would be effective in clarifying whether the observed phenotypic differences are genetically fixed or result from environmentally induced plasticity. In this study, we focused on phenotypic evaluation and did not conduct genome-level analyses. However, we recognize that future molecular-level investigations will be essential to elucidate the relationship between phenotypic traits and genetic background.

Comments 9: While of speaking of the genetic/morphological variability and then running a test for significance on such data sets. One limitation is the sample size and regions (e.g., Koshikijima) had fewer populations, potentially skewing results. Is there any reason for this selection? This may be provided at any relevant section in the MS.

Response 9: The limited number of populations in the Koshikijima Islands is due to the naturally restricted distribution of the species in that region. The methods for selecting and defining populations in each region were based on our previous studies (Ando et al. 2015, 2016), and a concise description of these methods is provided in the present manuscript. Additional explanations have been included in the “4-1. Growth Survey” section.

Comments 10: It is ascertained that greenhouse growth conditions may not fully replicate native environments required for Caryopteris incana, possibly masking some adaptive traits. Why not or why the experiment was taken in field-based trial?

Response 10: In this study, greenhouse cultivation was adopted to minimize the influence of environmental factors and to enable comparison of all populations under uniform conditions. In particular, Caryopteris incana is susceptible to poor growth under prolonged rainfall, making stable evaluation in outdoor settings difficult. Therefore, the greenhouse environment was considered more suitable. Additionally, due to space limitations at the research facility, outdoor cultivation was not feasible, and greenhouse cultivation was the most practical option.

Comments 11: The overall strength of the study is that, this study provides a baseline data/study for climatic resilience in Caryopteris incana. Still no climatic variables (e.g., precipitation, temperature) are correlated with phenotypic traits to predict responses to climate change. This may be overseen and if such data is available with the authors, appropriate details may be mentioned in the MS at any relevant section.

Response 11: This study primarily aimed to clarify phenotypic differences among regions; therefore, correlation analyses with climatic variables (e.g., precipitation, temperature) were not conducted. However, we recognize that investigating the relationship between climatic factors and phenotypic traits is an important subject for future research.

Comments 12: There is no way forward/recommendations based on this study and may be added.

Response 12: Based on your suggestion, we have added a statement at the end of the Discussion section to address the implications of our findings and future research directions.

Reviewer 2 Report (New Reviewer)

Comments and Suggestions for Authors

This study suggests each individual from the native population exhibited a variety of phenotypes by trait class, and the variation across and within regions indicated phenotypes specific to each region. However, there are still minor problems that need to be modified.

Q1: L30-36: These two sentences are repeated.

Q2: Paragraphs 1 to 3 seem to be introducing the characteristics of C. incana. The author should focus more on the research progress of this species

Q 3: 3-1. Traits related to phenotypic variation: This part contains too much data, much like the writing approach in the results section. The author should conduct a scientific discussion based on the result data of this article.

Q 4: The article needs to add a conclusion section.

Q 5: This article has relatively few references and most of them are rather outdated. Therefore, some viewpoints may be subjective and speculative. It is suggested that the author add some references, especially the research progress in recent years

Author Response

Comments 1: L30-36: These two sentences are repeated.

Response 1: The duplicated sentence in lines 30–36 has been removed, and the description has been revised for clarity.

Comments 2: Paragraphs 1 to 3 seem to be introducing the characteristics of C. incana. The author should focus more on the research progress of this species.

Response 2: At the beginning of the third paragraph of the introduction, we have briefly added a summary of previous research developments and the positioning of the present study to improve the overall balance of the description.

Comments 3: 3-1. Traits related to phenotypic variation: This part contains too much data, much like the writing approach in the results section. The author should conduct a scientific discussion based on the result data of this article.

Response 3: In response, we have added a clarifying statement at the beginning of the second paragraph of Section 3-1 to emphasize that this section is intended to provide scientific discussion based on the results, rather than a repetition of the data.

Comments 4: The article needs to add a conclusion section.

Response 4: At the end of the “Discussion” section, we have added a concluding summary, along with recommendations based on the findings of this study and suggestions for future research directions.

Comments 5: This article has relatively few references and most of them are rather outdated. Therefore, some viewpoints may be subjective and speculative. It is suggested that the author add some references, especially the research progress in recent years.

Response 5: In response, we have reviewed recent literature and added relevant citations to support our interpretations. Specifically, we have incorporated a recent study [9] that emphasizes the ecological significance of phenotypic diversity in plant adaptation.

Reviewer 3 Report (New Reviewer)

Comments and Suggestions for Authors

Overall statement:

The manuscript conducted a study to assess the phenotypic variation of C. incana in Western Kyushu and investigate its relationship with the native environment and geographical structure under the same growing conditions using collected seeds, these findings can give directions for understanding environmental adaptation of the plant species, and the information obtained on phenotypic variations could help differentiate the usefulness of plant resources. In addition, the research and the manuscript are relative well executed, on the basis of which I recommend its publication in Plants. Before fully accepted, I believe there are some points that need to be improved before fully accepted.

My main suggestions for improvement are:

   Because the relationship of trait variation and environment is object of this paper, i suggest that comparison statistic should be conducted to determine the phenotypic differences of the four areas. Some tables may be changed to figures for more clear showing results.

Some details are as following:   

Line 35, the words is repeated.  

Line 33, figure 1 is not necessary here, i suggest it can be combine with figure 2.

Line 95, figure 2, “Names of each region and locations of native populations”, the locations is not clear

Lines 128-130, “number of days in the flowering period” is not mentioned in the text.

Lines 452-470, the references is limited, more relative papers should be cited in the introduction and discussion sections.

Line x. All the tables and figures should be added more note information for their independence and accuracy.

Author Response

Comments 1: Because the relationship of trait variation and environment is object of this paper, I suggest that comparison statistic should be conducted to determine the phenotypic differences of the four areas. Some tables may be changed to figures for more clear showing results.

Response 1: We would like to clarify that our study employed two complementary approaches to compare phenotypic differences across the four regions. First, Table 1 presents the mean values of each trait by region, allowing for a direct comparison of phenotypic averages. Second, to provide a more comprehensive and interpretable overview of regional variation, we conducted principal component analysis (PCA) for each trait category. This dual approach enabled us to capture both specific trait-level differences and broader patterns of phenotypic divergence among regions.

Comments 2: Line 35, the words is repeated.  

Response 2: The duplicated sentence in lines 30–36 has been removed, and the description has been revised for clarity.

Comments 3: Line 33, figure 1 is not necessary here, I suggest it can be combine with figure 2.

Response 3: Thank you for your suggestion regarding Figure 1. We understand your point; however, we decided to retain Figure 1 because Caryopteris incana is not a widely recognized species, and we believe that presenting its morphology at the beginning of a study focused on phenotypic variation helps readers better understand the context. Additionally, Figures 1 and 2 have already been revised in response to other comments. We have also revised the captions of both figures to improve clarity and ensure that their content is more effectively communicated.

Comments 4: Line 95, figure 2, “Names of each region and locations of native populations”, the locations is not clear.

Response 4: We have revised the figure to improve the clarity of the geographic information. Specifically, we added dotted outlines to clearly indicate the boundaries of each region. We hope this modification enhances the readability of the figure.

Comments 5: Lines 128-130, “number of days in the flowering period” is not mentioned in the text.

Response 5: Thank you for pointing this out. We have added a description of how the number of days in the flowering period was measured and its variation across individuals and populations in the relevant section of the manuscript.

Comments 6: Lines 452-470, the references is limited, more relative papers should be cited in the introduction and discussion sections.

Response 6: In response, we have reviewed recent literature and added relevant citations to support our interpretations. Specifically, we have incorporated a recent study [9] that emphasizes the ecological significance of phenotypic diversity in plant adaptation.

Comments 7: Line x. All the tables and figures should be added more note information for their independence and accuracy.

Response 7: In response, we added annotation information to selected figures and tables to enhance their clarity and independence. Some of these modifications were also made in response to comments from other reviewers, and we carefully revised the captions to ensure accuracy and consistency.

Reviewer 4 Report (New Reviewer)

Comments and Suggestions for Authors

Dear authors, congratulations on your work.

This manuscript provides a comprehensive evaluation of phenotypic variation in Caryopteris incana across Western Kyushu. The study addresses an important gap in knowledge regarding intraspecific diversity in this threatened species, using a common garden experiment to isolate phenotypic differences across regions. The work has merit and potential for publication but requires clarification and revision in several key areas to improve scientific accuracy, clarity, and interpretability.

Major comments:

Introduction

  1. Conservation Status – Line 69

The manuscript refers to C. incana as listed as “Vulnerable II (VU)” in the Red Data Book. This is ambiguous and potentially misleading. Please clarify terminology, use  “Vulnerable (VU)” per IUCN guideline ana explain if the “II” refers to a national subcategory (if applicable).

  1. Figure 2 – Native Populations Missing

Panel B of Figure 2 only shows sampled populations. Given that 109 populations were identified, it would be informative to display all known native populations, either in the same map or an additional panel.

  1. Table 1 – Legend and Presentation

The current legend does not explain the meaning of superscript letters (a, b, c), and not all traits include standard error (SE). Expand the legend to clarify that superscript letters indicate statistically significant differences (P<0.05) between regions. Also, include SE for all mean values for transparency.

  1. Color Traits – Explanation of L, a, b

The meaning of L, a, and b values for flower and leaf color is not explained. Add a short description of the CIE Lab color space and explain what each value represents.

Results

  1. KI Traits Not Fully Discussed. Figure 3 suggests that KI populations have not only larger floral organs but also brighter flowers. This is not adequately emphasized in the discussion. Include a sentence clarifying that KI flowers are both larger and brighter and speculate on ecological relevance.

Discussion – Conservation Implications

  1. The section discussing conservation value (Section 3.4) lacks a strong connection to how phenotypic variation informs conservation prioritization.

Methods – Experimental Conditions and Rationale

  1. It is unclear why the authors chose to grow plants under greenhouse conditions rather than assess traits in situ (e.g., to reduce environmental variability). Please justify the use of controlled cultivation.
  2. Sampling Design – Population Numbers

The number of populations sampled per region is only partially detailed in the Methods. A complete overview is missing. Moreover, it is recommended to include a table listing the number of populations sampled per region and how many individuals were assessed per trait.

  1. Measurement Standardization – Internode and Serration

The use of the “10th internode” and “5th serration” needs explanation. Please, add a rationale that these points were selected to standardize leaf age and maturity across samples.

  1. Incorrect Terminology – Line 394

The term “male and female stamens” is incorrect. Stamens are male structures; pistils are female. Please, replace with “stamens and pistils” or “male (stamens) and female (pistils) floral organs”.

Minor Comments

  1. Figures 3–5: Clarify in captions what PC1 and PC2 represent in terms of trait categories. Define TI, NM, GI, KI at first mention and consider a table of abbreviations.

  1. English clarity: Some expressions (e.g., “number of flowering side branches”) can be rephrased for fluency.
  2. Statistical methods: Specify in the text which post-hoc test was used per trait if variance assumptions varied.

I appreciate your work on this important topic and look forward to seeing the final version.

Author Response

Comments 1: Conservation Status – Line 69; The manuscript refers to C. incana as listed as “Vulnerable II (VU)” in the Red Data Book. This is ambiguous and potentially misleading. Please clarify terminology, use “Vulnerable (VU)” per IUCN guideline ana explain if the “II” refers to a national subcategory (if applicable).

Response 1: We agree that the expression "VU II" may be ambiguous. We have revised the manuscript to clarify that "Vulnerable (VU)" follows the IUCN Red List category, and "II" refers to the national classification used in Japan’s Red Data Book.

Comments 2: Figure 2 – Native Populations Missing; Panel B of Figure 2 only shows sampled populations. Given that 109 populations were identified, it would be informative to display all known native populations, either in the same map or an additional panel.

Response 2: In our previous study, we identified 109 wild populations of Caryopteris incana. However, in the present study, only the populations from Tsushima were selected for detailed analysis. Therefore, all known wild populations relevant to this study are already represented in Figure 2B. 

Comments 3: Table 1 – Legend and Presentation; The current legend does not explain the meaning of superscript letters (a, b, c), and not all traits include standard error (SE). Expand the legend to clarify that superscript letters indicate statistically significant differences (P<0.05) between regions. Also, include SE for all mean values for transparency.

Response 3: As noted in the caption of Table 1, the superscript letters (a, b, c) indicate statistically significant differences among regions (P < 0.05). Regarding the standard errors (SE), we have omitted them from the table to maintain clarity due to the large number of traits. Instead, we discuss trait variability using coefficients of variation in Table 5, which provides a comprehensive overview of variation across and within regions.

Comments 4: Color Traits – Explanation of L, a, b; The meaning of L, a, and b values for flower and leaf color is not explained. Add a short description of the CIE Lab color space and explain what each value represents.

Response 4: To clarify the meaning of the L, a, and b values used for flower and leaf color traits, we have added a brief explanation of the CIE Lab color space in the relevant section of "4-2. Measurement method".

Comments 5: KI Traits Not Fully Discussed. Figure 3 suggests that KI populations have not only larger floral organs but also brighter flowers. This is not adequately emphasized in the discussion. Include a sentence clarifying that KI flowers are both larger and brighter and speculate on ecological relevance.

Response 5: We have clarified in Section 3-3. that the KI populations are characterized by large floral organs and bright coloration. Additionally, we have added a description explaining that this bright coloration is based on high L and b values. Furthermore, we briefly discussed the potential ecological significance of this trait combination, suggesting that it may be associated with delayed and synchronized flowering in the KI region and may contribute to enhanced pollinator attraction.

Comments 6: The section discussing conservation value (Section 3.4) lacks a strong connection to how phenotypic variation informs conservation prioritization.

Response 6: We have revised Section 3.4 to more clearly articulate how phenotypic variation informs conservation priorities. Specifically, we emphasized the importance of preserving populations with unique or extreme traits, as these may represent valuable genetic resources for both ecological adaptation and horticultural breeding.

Comments 7: It is unclear why the authors chose to grow plants under greenhouse conditions rather than assess traits in situ (e.g., to reduce environmental variability). Please justify the use of controlled cultivation.

Response 7: In this study, greenhouse cultivation was adopted to minimize the influence of environmental factors and to enable comparison of all populations under uniform conditions. In particular, Caryopteris incana is susceptible to poor growth under prolonged rainfall, making stable evaluation in outdoor settings difficult. Therefore, the greenhouse environment was considered more suitable. Additionally, due to space limitations at the research facility, outdoor cultivation was not feasible, and greenhouse cultivation was the most practical option.

Comments 8: Sampling Design – Population Numbers; The number of populations sampled per region is only partially detailed in the Methods. A complete overview is missing. Moreover, it is recommended to include a table listing the number of populations sampled per region and how many individuals were assessed per trait.

Response 8: Instead of presenting the information in a table, we have addressed your suggestion by adding a descriptive sentence in the main text. Specifically, we included this information at the end of section 4-2 to clarify the number of populations surveyed in each region and the number of individuals evaluated per trait category.

Comments 9: Measurement Standardization – Internode and Serration; The use of the “10th internode” and “5th serration” needs explanation. Please, add a rationale that these points were selected to standardize leaf age and maturity across samples.

Response 9: We have added an explanation to clarify that the 10th internode was chosen to ensure leaf maturity and reduce developmental variation, and that the 5th serration was used as a consistent reference point for standardizing measurements across individuals.

Comments 10: Incorrect Terminology – Line 394; The term “male and female stamens” is incorrect. Stamens are male structures; pistils are female. Please, replace with “stamens and pistils” or “male (stamens) and female (pistils) floral organs”.

Response 10: We have corrected the phrase “male and female stamens” to “stamens and pistils” to accurately reflect floral anatomy.

Comments 11: Figures 3–5: Clarify in captions what PC1 and PC2 represent in terms of trait categories. Define TI, NM, GI, KI at first mention and consider a table of abbreviations.

Response 11: In response, we have added clarifications to the captions of Figures 3 to 5 to indicate which trait categories are represented by PC1 and PC2. Regarding the abbreviations TI, NM, GI, and KI, we have stated at their first appearance in section "2-1. A) Phenology" that these are used as abbreviations for each region.

Comments 12: English clarity: Some expressions (e.g., “number of flowering side branches”) can be rephrased for fluency.

Response 12: We have changed "number of flowering side branches" to "number of lateral flowering branches".

Comments 13: Statistical methods: Specify in the text which post-hoc test was used per trait if variance assumptions varied.

Response 13: As described in section "4-3. Statistical Analysis", we have specified the post-hoc tests used for each trait category. The selection of post-hoc tests was made based on the assumptions of variance equality, ensuring statistical rigor under each condition.

Round 2

Reviewer 2 Report (New Reviewer)

Comments and Suggestions for Authors

The author's revisions have greatly improved this article. However, in the introduction and discussion sections, more references to the latest literature are still needed as support

Author Response

Comments 1: The author's revisions have greatly improved this article. However, in the introduction and discussion sections, more references to the latest literature are still needed as support

Response 1: Thank you for your valuable comment. In response, we have added a citation to [11]Wang et al. (2023) in the Discussion section (3-4. Usefulness as a plant resource and conservation) to emphasize the relevance of phenotypic diversity in enhancing resilience to environmental disturbances. This addition supports the conservation significance of the observed intraspecific variation in Caryopteris incana.

Reviewer 4 Report (New Reviewer)

Comments and Suggestions for Authors

Dear authors, 

Thank you for your detailed responses. I appreciate the clarifications provided. However, I would still recommend the following revisions to improve clarity and completeness:

Figure 2: Thank you for the explanation. However, to enhance the completeness and geographic context of Figure 2, I recommend indicating the locations of all previously identified wild populations of Caryopteris incana, even if only a subset was analyzed in this study. Including all known native populations—either in the same figure or in an additional panel—would provide valuable context regarding the species' overall distribution and help readers understand the sampling scope more clearly.

Table 1: Thank you for the clarification. However, for full transparency and clarity to the reader, I recommend explicitly stating in the legend of Table 1 that superscript letters (a, b, c) indicate statistically significant differences among regions (P < 0.05). Additionally, it would be helpful to include a brief note in the legend referring the reader to Table 5 for trait variability, since standard errors are not provided.

Author Response

Comments 1: Figure 2: Thank you for the explanation. However, to enhance the completeness and geographic context of Figure 2, I recommend indicating the locations of all previously identified wild populations of Caryopteris incana, even if only a subset was analyzed in this study. Including all known native populations—either in the same figure or in an additional panel—would provide valuable context regarding the species' overall distribution and help readers understand the sampling scope more clearly.

Response 1: Thank you for your constructive feedback. In response, we have added a new panel (Figure 2C-F) that provides region-specific distribution maps for all known wild populations of Caryopteris incana in Tsushima (TI), Nagasaki Mainland (NM), Goto Islands (GI), and Koshikijima Islands (KI). In the Tsushima region, 72 populations have been recorded, of which 20 populations used in this study are highlighted in red.

Comments 2: Table 1: Thank you for the clarification. However, for full transparency and clarity to the reader, I recommend explicitly stating in the legend of Table 1 that superscript letters (a, b, c) indicate statistically significant differences among regions (P < 0.05). Additionally, it would be helpful to include a brief note in the legend referring the reader to Table 5 for trait variability, since standard errors are not provided.

Response 2: In response, we have revised the legend of Table 1 to explicitly state that superscript letters (a, b, c) indicate statistically significant differences among regions (P < 0.05). Additionally, we have added a note referring readers to Table 5 for trait variability, as standard errors are not shown in Table 1.

This manuscript is a resubmission of an earlier submission. The following is a list of the peer review reports and author responses from that submission.

Round 1

Reviewer 1 Report

Comments and Suggestions for Authors

My comments on the manuscript are as follow:

1.     I have read the manuscript now, the paper is important and may of interest to a wider community and therefore worth reporting. The authors have carried out extensive collection/surveys for Caryopteris incana an endangered species distributed within the western Kyushu area of Japan. Further, the authors have identified 72 populations on Tsushima, Nagasaki Prefecture, 16 populations on the mainland of Nagasaki, 16 populations on the Goto Islands, and 109 populations on Koshikijima Islands, Kagoshima Prefecture with morphometric variation that is reassessed with chloroplast DNA/gene markers that was previously done [Ref. 8]. On the brighter side, the text is clear and conveys a clear message to the readers. The study has attempted to represent/explain a wider picture of the available morphotypes within the area and stress is given for conservation of these variations/types for future.

2.     On the contrary, the manuscript is structured in a way that is making it unclear as to what are the novel results and what is known earlier? For instance there is nothing/or vey limited information till the end of the MS that the DNA based studies are not done by the authors here, rather they have referred/compared their results with the previously data. Also how may representative samples were taken/assessed for the DNA based studies and how are those samples related.

3.     Similarly, objectives laid down are not synchronized with the outcomes of the study and are equally not reflected in the title. The title is dealing with morphometric evaluation whereas the abstract included DNA based (haplotypic) analyses. Later in the Introduction section for example, “……….The objective of this study was to investigate phenotypic variation in each natural population by growth survey and to evaluate its association with geographical structure. These results suggest that the phenotypic diversity of the indigenous population in western Kyushu and comparison with the genetic structure in chloroplast DNA may clarify the relationship between phenotypic variation and genetic variation. Information on these phenotypes may contribute to the utilization of plant resources such as breeding materials and the planning of conservation programs as endangered species ………….”(L91-95). I do not see any scientific justification for this within the text as there are no enough resulted to justify such claims.

4.     The morphological variability is thoroughly elaborated and is enough to be reported, however if the DNA based data is to be included it must be clearly mentioned with a logic given and the proper source/citation is to be given at start along permission form the American journal of Plant sciences (if it is not taken yet, and even if it is the authors own data that is published earlier). Since the morphological data sets/results are not fully integrated within the molecular and also as the molecular data is not having the full representation of the area sampled (unless I have missed it). I would suggest to see a justification for that in the rebuttal.

5.     Who is followed for these recordings/calculations of morphological attributes i.e. any descriptor used, citation of that may be provided. Neither I see who has authorised this study, as it is an endangered plant and is collected form a vast geography and if any specimens of these morphotypes are submitted to any organization as voucher specimen?

6.     My serious concern is, that except for the text there is not enough correlation of the DNA haplotypes related data with the morphotypes. This needs to be done in some sorts of analyses and graphical portraying will make that easy to interpret for the readers (the authors themselves have used multivariate tools and I will highly recommend similar approach to be followed for this composite analyses)

7.     A good part of the introduction section L65-67 could be shifted to the Methodology section; whereas L73-85 may be shifted to Results or Discussion where ever it is more justified.  

There are few minor issues and may be addressed:

a.     There are typos and grammar related issues, specifically when elongated statements are used e.g.

b.     Any abbreviations used needs to be described in full at their first place of mention.

c.     L25: C.×clandonensis is without space and needs correction. Is it mandatory to marks the “x” symbol after its first use. Otherwise “hybrids” may be used instead.  

d.     L49-50: “…….exerts potent insecticidal effects……” may be rephrases as “having potential insecticidal properties……” or any other way to provide a clear message.

e.     References needs to be re-assessed for consistently and followed as per the journal format/standards.

Decision:

While the study is within the scope of the journal, and information may be handy and of wider interest. The MS in the current form cannot be accepted, it needs Revisions.

Comments on the Quality of English Language

Nil

Reviewer 2 Report

Comments and Suggestions for Authors

Reviewer's comments

Article Plants – 2602627

This manuscript addresses a timely topic and makes an important contribution to the field of intraspecific phenotype variation of Caryopteris incana in Japan. However, it needs some major revisions before it can be published. The manuscript is based on impressive empirical evidence and represents an original contribution. The authors work with an interesting data set. However, the statistical analysis is inadequate, also and the presentation of analyzes. I believe that some additional corrections can improve the authors’ argument.

As it stands, this work cannot be considered for publication. However, I see the value of the research approach and encourage the authors to revise and resubmit their manuscript.

Major comments

-The authors need to revise their manuscript to show intraspecific variation in the phenotype of Caryopteris incana for different categories of plant traits (see below). According to the results of the statistical analyzes, discuss the within-site and between-site variation and explain the observed patterns of variation for different trait categories.

-Organize the analyzes, results, and discussion by trait category: Phenology (days to flowering and flowering time), Vegetative (leaf and plant traits), Reproductive (flowering traits), Reproductive fitness traits (fruiting/number of inflorescences, plant height at fruiting and spike length/(number of inflorescences), DNA analysis results (chloroplast DNA haplotypes).

-First, the mean values of the traits for all four studied sites (Tsushima, Nagasaki, Goto and Koshikijima Islands) must be presented, e.g. histogram of mean values - average values of the studied populations of each site by trait categories.

The statistics and presentation of results must be repeated in an appropriate manner.

In ANOVA analyzes, include the source of phenotype variation in the model:

Site

(statistical significance confirms differences caused by environmental conditions at sites - interspecific variation)

Population

(nested within sites) (statistical significance confirms differences between populations within sites, differences in environmental conditions between different populations – interpopulation variation)

-If all colors are represented in populations, add flower color factor nested within site and population. The results confirm the differences between colors within sites and populations, the interpopulation variation. In this way, we obtain a result indicating whether there is phenotypic variability between colors.

-In the model of ANOVA, sites and color are a fixed factor and populations are random.

-A post-hoc test was used to determine the differences between the means of the analyzed traits.

-PCA analyzes as multivariate analyzes must include the original values of the traits, not the ratios or derived traits. Repeat Analyzes. Scale down and to the left in the figure, not to the center of the figure. Use only the PC1 and PC2 scatter plots.

-Multivariate analyzes such as principal component analysis (PCA) examine the main patterns of variation in a data set. Determine the differences among the studied sites and evaluate the contribution of individual plant traits using canonical discriminant analysis (CDA).

-Present the cluster analysis in the form of a dendogram.

- Not enough references

Other suggestions:

Add an illustration with flower color.

Explain

The color of the leaf presents what ????

Flower shape present ????

Plant shape ????

Page 2

Line 68

Figure 1.

The (A) part of Figure 1 is reduced in size compared to the (B) part. Add some markers (e.g., circles) on the (B) part. Add the labels Tsushima, Nagasaki, Goto and Koshikijima and use the size of the markers according to the number of observed populations.

Line 76

Plot the results of interpopulation variation of chloroplast DNA in the figure.

Page 6

Line 152

Figure 3

Use different colors for each of the phenotypic groups.

Reviewer 3 Report

Comments and Suggestions for Authors

Dear authors,

I had an opportunity to review the article titled "Intraspecific evaluation of phenotypic variation of Caryopteris incana in western Kyushu, Japan".

The authors investigated phenotypic variation of Caryopteris incana, plant species threatened in Japan, and of horticultural interest. They also tried to combine these results form those from their previous studies in order to detect population of conservation concern.

My detailed comments can be found in the pdf version of the manuscript. Here, I will emphasize just a few things.

1. The English language must be thoroughly reviewed by an appropriate English speaking person, as there are many parts of the text that are not clear because of the mistakes in the English language. 

2. Figure 1 must be changed or adequately cited as it is completely the same (including the legend) as in the article Ando et al. (2016): American Journal of Plant Sciences 7: 167-180.

3. Concerning the results, in the Supplementary material 1 the authors provide only average values for the measured parameters, without any basic statistics. Since this is primarily the analysis of morphological characteristics, SD, Min and Max could be quite useful.

4. Figure 3 must be improved, as in the present form it is almost not useful. It has to be larger, with colored symbols.

5. The results of the correlation analyses must be presented as a table in the manuscript, or in the Supplementary material.

6. The discussion is a little bit strange, as the authors present some descriptive conclusions without scientific back up.

For example, they say that there is no correlation between phenotypes and environmental conditions, but this analysis is not at all present in the manuscript (and it would be a huge plus to have it).

Comparison of phenotypes and haplotypes is just descriptive. These two types of the results should be statistically/graphically overlapped, to really see the situation between haplotypes and phenotypes. Ideally, the molecular and morphological analysis should have been done on the same individuals.

The authors determined populations of high conservation priorities mostly based on the number of individuals in the populations and their haplotypes, and much less on their phenotype. Anyway, I think it would be useful to have some table with overall results for every population combining the number of individuals, haplotype, and phenotype.

The authors should try finding and comparing their results with some similar studies on threatened plant species, where there was discrepancy between phenotype and haplotype in the threatened plant species.

7. Methodologically, the results are not backed up by adequate sample size to be published in the renowned journal as Plants.

We do not know from how many individuals seeds were originally collected.

The measurements of morphological characteristics were done on just 5-10 individuals per populations, which is very low for this kind of research.

The real analysis correlating phenotype, haplotype, and environmental characteristics is missing.

Based on all written above, I cannot support publication of the presented results in "Plants".

With my kindest regards,

Comments on the Quality of English Language

The English language must be thoroughly reviewed throughout the text by an appropriate English speaking person, as there are many parts of the text that are not clear because of the mistakes in the English language.